# Qualitative Evaluation of the STOEMP Network in Ghent: An Intersectoral Approach to Make Healthy and Sustainable Food Available to All

**DOI:** 10.3390/ijerph17093073

**Published:** 2020-04-28

**Authors:** Marjolijn Vos, Maria Romeo-Velilla, Ingrid Stegeman, Ruth Bell, Nina van der Vliet, Wendy Van Lippevelde

**Affiliations:** 1Department of Marketing, Innovation and Organisation, Faculty of Economics and Business Administration, Ghent University, 9000 Ghent, Belgium; Marjolijn.Vos@ugent.be; 2Flemish Institute of Healthy Living, 1000 Brussels, Belgium; 3EuroHealthNet, 1040 Brussels, Belgium; m.romeo-velilla@eurohealthnet.eu (M.R.-V.); i.stegeman@eurohealthnet.eu (I.S.); 4Institute of Health Equity, University College London, London WC1E 7HB, UK; r.bell@ucl.ac.uk; 5National Institute for Public Health and the Environment (RIVM), Centre for Sustainability, Environment and Health, 3720 BA Bilthoven, The Netherlands; nina.van.der.vliet@rivm.nl; 6Department of Nutrition and Public Health, Faculty of Health and Sport Sciences, University of Agder, 4604 Kristiansand, Norway

**Keywords:** INHERIT, STOEMP, Ghent, food policy, network, interviews, health, sustainability, equity, systems approach

## Abstract

The STOEMP network is, to our knowledge, one of the first initiatives to bring different sectors together in a municipality so as to increase accessibility to healthy and sustainable foods for all, with particular attention for the disadvantaged population. This qualitative study aimed to gain an in-depth insight into how the STOEMP network aims to reach its goal of making healthy, sustainable food available to everyone, through an intersectoral, collaborative process, exploring the facilitators and challenges of taking a systems-oriented approach to achieving this. Interviews were conducted among 15 stakeholders of the STOEMP network between March–July 2019 in Ghent (Belgium). Factors that facilitated the development and work of the network are reported, including having an external, neutral process manager, shared values, multisector engagement, enthusiasm, resources, and sense of ownership, as well as the barriers that were faced, such as time issues, uncertainty regarding continuation and funding, and discrepancy in visions. These issues reflect the strengths and challenges of taking a systems approach that aims to formulate solutions to widening access to healthy and sustainable foods. STOEMP would like to influence policy and thereby strengthen its impact, but needs further discussions to collectively formulate exact needs.

## 1. Introduction

Global health problems such as obesity, diabetes, and other non-communicable diseases have been on the rise in the last decades [1,2]. Additionally, current food consumption and production patterns (i.e., a high intake of animal-based foods such as meat and dairy products) damage our ecosystems at a global scale, and contribute to the depletion of natural resources and climate change [3]. Thus, there is an urgent need to transition to healthier and more sustainable food systems that promote nutrient-dense plant-based food consumption patterns [4]. However, changing food consumption behavior is challenging due to its complex nature, for example many consumption behaviors including meat eating are habitual with relatively little conscious reflection [5,6]. Moreover, many environmental factors influence consumption behavior, such as the availability and affordability of certain foods or social influence of peers and family [6,7,8,9]. “A transformation of the food system should ultimately involve multiple stakeholders, from individual consumers to policy makers and all actors in the food supply chain, working together towards the shared goal of healthy and sustainable diets for all.” [2,4].

Given the existing social gradient in health (i.e., individuals with a lower socioeconomic position have less healthy lifestyles, and more health problems than individuals with a higher socioeconomic position), and because those with a lower socioeconomic position are more likely to be adversely affected by the effects of climate change, it is of crucial importance to promote healthy and sustainable diets for all, especially amongst the less and least advantaged in society [10,11,12]. However, earlier research showed that most population-based health promotion efforts (e.g., mass media social marketing initiatives, front-of-pack labelling) can widen social inequalities in health [13,14]. Many such interventions (e.g., mass media social marketing initiatives) include strategies that rely on individuals’ cognitive, psychological, time, and material resources, which tend to be socioeconomically patterned [15]. Adams and colleagues [15] highlighted that population-based interventions that require individuals to use a low level of agency (i.e., fewer personal resources) to benefit are likely to be most effective and equitable. Moreover, local strategies or policies aimed at structural changes to the environment have been shown to be promising, especially for more disadvantaged populations [13]. Setting-based approaches in cities and towns provide opportunities to implement policies and interventions to make healthy and sustainable food available to all.

The WHO’s European Healthy Cities Network (EHCN) is one of the most well-known setting-based approaches to implement health promotion and increase health equity. The EHCN has brought together some 100 flagship cities and approximately 30 national networks to promote EHCN’s strategic priorities of health equity, partnership, and health in all policies [16]. Although the EHCN evaluation showed a contribution to urban health development with health equity as one of its key values [16], Ritsatakis [17] found that when it came to interventions specifically designed to improve population health, cities mainly implemented downstream actions (i.e., individual-level behavioural interventions). Few cities made the shift to more upstream policies (i.e., policy approaches that can affect large populations through structural changes e.g., regulation, changes in access, or economic incentives) to tackle health determinants. Other systematic reviews confirmed that cities tend to implement more downstream actions [18,19]. These reviews [18 and 19] also acknowledged that more effective initiatives to reduce health inequalities require less focus on individual behavior change interventions within settings and more on interventions which change the structure of settings themselves. For example, previous evidence showed that accessibility is a crucial determinant of dietary intake that can facilitate or hinder healthy (and sustainable) eating [20]. Furthermore, Pons-Vigues and colleagues [19] argued, based on several reviews and WHO reports [21,22,23,24], that to tackle the inequalities in health an intersectoral approach that addresses social and environmental inequalities was required and essential, and more successful than action taken solely by the health care sector. However, their analysis found that most of the policies in European cities still applied a fragmented and sectoral approach [19]. Some cities are now applying a strategic whole of city government approach to tackling health inequalities and inequalities in social determinants, including Coventry and Greater Manchester in England [25].

The current study was conducted in the framework of the H2020 “Inter-sectoral Health and Environment Research for Innovation” (INHERIT) project (2016–2019) that aimed to identify effective inter-sectoral policies, interventions, and innovations that enable a ‘’triple win’’: To reduce environmental impacts, improve health and wellbeing, as well as generate greater health equity. INHERIT identified promising examples of triple-win practices, so as to investigate how they can be multiplied and scaled, as a contribution to addressing inter-related environmental, health, and social challenges that societies are facing [26]. Since interventions that effectively address all dimensions of the ‘triple-win’ require cooperation across sectors, INHERIT’s work placed a strong emphasis on investigating facilitators, as well as barriers to cooperation, and how these might be overcome. INHERIT also put a strong emphasis on behavior change as insights from the field of behavior change are not sufficiently considered or exploited in efforts to address current environmental, health, and social challenges. INHERIT’s work was underpinned by a Conceptual Model developed at the outset of the project, that brought together the key themes that it was working with. Nested within the Model is the ‘Behavior Change Wheel’ [27] which recognizes the roots of behavioral change in the Capability, Opportunity, and Motivation (COM-B) of people, and usefully relates these to different kinds of actions and policies that can be implemented to modify the COM-B factors.

The STOEMP network was identified by INHERIT as a ‘‘promising practice’’, that is, an intervention with strong potential to achieve a ‘triple-win’. The Ghent en Garde Food Policy in which STOEMP sits is, to our knowledge, one of the first city food policies that has put a specific emphasis on how to make healthy and sustainable foods available to all, including more disadvantaged populations. The STOEMP network was brought together as a cooperation between different sectors in the municipality to explore the question of what can be done to achieve this objective. The aim of this study, which took place in the broader context of the INHERIT project, was to capture insights into how the process of intersectoral cooperation takes place within STOEMP. The aim was not to identify which approaches and policies have been identified to make ‘good’ food available to disadvantaged populations, since these have not been explicitly formulated at the time of research. It was, rather, to investigate the process of intersectoral cooperation, and the factors that facilitated this (providing participants with the capabilities, opportunities, and motivation to do so), as well as the barriers involved in bringing different actors from different organizations and sectors together to identify what is needed to make healthy, sustainable food available to everyone. The findings can inform and facilitate the creation of similar approaches and networks in other municipalities.

## 2. Method

### 2.1. Description of the STOEMP Network

The Belgian city of Ghent is a forerunner in the promotion of sustainable food on a city-wide level and has developed a comprehensive food policy, called ‘‘Ghent en Garde’’, which was launched in 2013. The food policy includes five strategic goals to pave the way for a sustainable food system for Ghent: (1) A shorter, more visible food chain; (2) more sustainable food production and consumption; (3) the creation of more social-added value for food initiatives; (4) reduce food waste; and (5) optimum reuse of food waste as raw materials. These goals were agreed upon following stakeholder discussions, input from the city administration, and political agreement. ‘’Ghent en Garde’’ entails several local, sustainable food initiatives, of which STOEMP is one.

STOEMP was launched in 2016 as a network of initiatives that aim to take forward the concept of sustainably-produced, healthy food and to make sure that it is made more widely available to vulnerable populations in Ghent. As with the city’s overall ‘‘Ghent en Garde’’ food policy, the focus has been on improving the status quo when it comes to the production and consumption of more healthy and sustainable food, rather than on pursuing rigorous definitions of what this is. STOEMP has therefore established itself as a network that aims to promote ‘’good food’’, which it defines as healthy, nutritious, and local, as well as produced with respect for the environment, sufficient for all, tasty, and fair. In Dutch, STOEMP means both a vegetable stew including a range of (mostly) healthy ingredients (indicating the multidisciplinary character of the team) and a push (indicating a ‘’nudge’’ or ‘’steer’’ in the right direction). This network includes stakeholders from different sectors and policy domains, including education, civil society, research, and social welfare. STOEMP focuses on three main objectives: (1) Raise awareness amongst policy makers, civil society, local economy, and the broad public about access to ‘’good food’’; (2) inspire and activate policy makers, civil society, local economy, and the broad public to work individually or together on providing ‘’good food’’; and (3) connect and strengthen initiatives to make ‘‘good food’’ available for everyone. The goals of the STOEMP network were agreed by a number of stakeholders, namely representatives of the municipality (i.e., Department Urban Planning and Welfare and Equal opportunities), community health centres and the local health promotion organization (LOGO+ Ghent). Since its start, several organizations have joined, most around its inception, and the network now is comprised of 25 organizations. The network’s main activities include three-monthly meetings open for everyone to join (not only the members of the network), and one inspiration event per year, including activities such as a visit to good practices in Ghent. A core group within the network organizes the open meetings and inspiration events. Additionally, STOEMP involves several thematic working groups that stakeholders can take part in namely: GoodFood@School; data collection; Ratatouille (i.e., a health education project for disadvantaged parents); policy impact; and communication.

### 2.2. Study Design and Sampling

A qualitative study including semi-structured interviews was conducted by the first and last authors of this paper (M.V., W.V.L.) between March and July 2019 among 15 stakeholders of the STOEMP network in Ghent, Belgium. Stakeholders were defined as representatives of organizations that are a member of the STOEMP network, covering different policy domains (environment, health, and social welfare) and sectors (policy, public, and private). There is no formal subscription commitment or fee, but rather interested organizations can join the network and add their organization’s representative to the mailing list. Representatives from organizations that work with disadvantaged groups are part of the STOEMP network. However, so far representatives of the target group (i.e., citizens of Ghent, with a particular focus on the disadvantaged groups) have not been included in the network.

The data collection was conducted in 3 waves (each including 5 participants). The first wave included interviews with the stakeholders that initiated the network in 2016. The second wave targeted stakeholders that joined at a later point in time, but have been intensively involved. The third wave included newer stakeholders as well as stakeholders from organizations and sectors that were involved in STOEMP even though its goals are not part of the STOEMP core task (i.e., restaurant, grocery store, and school). We also invited stakeholders that resigned during the previous years (*n* = 3) but none showed interest in participating. Purposive (heterogeneity or maximum variation) sampling was used for the two first waves based on the participant list of the STOEMP network received in early 2019 from the STOEMP coordinator, while a snowball technique was used to find the participants of the third wave based on suggestions from participating stakeholders. An overview of the different participating stakeholders and their professional contact details was provided by the STOEMP coordinator. The researchers sent emails to possible participants asking for their involvement in the study. The letter specified the study objectives and data collection procedures. In total 25 (ex-) stakeholders of STOEMP were invited, of which 16 participated. One interview was discarded due to the participant’s lack of knowledge about the STOEMP network. Reasons for non-participation were job changes, or lack of interest amongst resigned STOEMP members, time constraints of current members, or feelings of inexperience regarding STOEMP. Ethical approval was granted by the Ethical Committee of the University Hospital Ghent (Belgium).

### 2.3. Data Collection

Interviews were conducted in a quiet location at the organizations of the representatives involved or at the University of Ghent. To obtain standardization in procedures, a protocol was written and followed by the interviewer [28]. A semi-structured interview guide was developed using the principles of Appreciative Inquiry [29] and the COM-B model [27]. Appreciative inquiry is a strength-based approach to gather data that focuses on gaining insight into success factors and further developments in the future as an alternative to a problem-centered approach. Appreciative inquiry was successfully applied in interviews before [30]. The COM-B model, embedded in the INHERIT Model, is a framework that includes constructs of Capability, Opportunity, and Motivation to understand behavior [26,27]. The interview schedule covered the following topics: Initiation of the network, how the network evolved, factors facilitating (and hindering) (net)working, success stories in reaching vulnerable groups, and future plans. At the beginning of the interview, participants completed a questionnaire that asked for demographic information (age, gender), how long they had been involved in STOEMP, as well as the organization and sector they represented. Interviews lasted an average of one hour. Informed consents, which stated that study results would be handled anonymously and interviews would be audiotaped, were obtained from all participants. All interviews were transcribed verbatim by external transcribers.

### 2.4. Data Analysis

A multi-stage analytical process applying thematic analysis that combined deductive, inductive, and verification techniques to check the reliability and validity of the coding system was conducted using NVivo 11.0 (QSR International, Melbourne, Australia) [28]. Firstly, a code book was created a priori by M.V. and W.V.L., which was based on the principles of Appreciative Inquiry and the COM-B model [27,29]. Secondly, the transcripts were read as an initial exploration of the data by W.V.L. Thirdly, transcripts were imported into NVivo 11.0 and the code book was used to code the interviews (deductive analysis) in combination with an inductive reflection on the analysis (generation of themes) [31]. Table 1 gives an overview of the theoretically deduced codes and the inductively produced themes. In terms of reliability and validity, one third of the transcripts (*n* = 5) was coded by two researchers (M.V., W.V.L.), afterwards codes and results were compared to check if consistent results were reached (in the few cases results were not consistent, the two researchers reached consensus and this was applied to the data). The other two third of the transcripts (*n* = 10) was further coded by one researcher (W.V.L.). In order to add quotes in the result section of this manuscript, W.V.L. translated the quotes from Dutch to English.

## 3. Results

### 3.1. Characteristics of Participants

Table 2 shows the organization, sector, and length of time that study participants had been involved in STOEMP when the interviews took place. In general, most participants were female (*n* = 13), with a mean age of 36.6 ± 8.2. Participants came from a variety of organizations and sectors.

### 3.2. Initiation and Development of the Intersectoral STOEMP Network

The study participants of the core team (wave 1) that started STOEMP provided a comprehensive account of the initiation of this network, which was verified by those interviewed in later waves. It originated from the development of two different working groups. The Food Council of Ghent, coordinated by the Environment and Climate Unit and consisting of both profit and non-profit organizations, is responsible for the local food policy. This council decided that it wanted to invest further in one of their strategic goals, namely creating more social value. To make this happen, several stakeholders of the Food Council brainstormed and subsequently aimed to target access to affordable, sustainable, and healthy food among disadvantaged groups. At the same time, the Department of Welfare and Equal Opportunities of Ghent conducted their yearly assessment of their work with Community Health Centres and the Local Health promotion organization. Based on these evaluations, the parties involved concluded that despite all efforts to reach vulnerable groups and promote health, it was vital to think of a better way to create more impact and enable health for all. Food was seen as one of the entry points for this approach. As soon as these two working groups heard of each other’s plans they decided to merge and that was the start of the STOEMP (net)working group. Several municipal councilors that were involved in this creation phase announced that there was some budget that could be allocated to start up this project.

The next step of STOEMP entailed the decision about what this working group would actually do. All interview participants highlighted the difficulty as well as necessity of this phase to define the mission and goals. Several participants underlined that it took time to build trust between the different sectors and organizations as well as to fine-tune the mission and goals. However, they all agreed that the involvement of an external “neutral” process supervisor facilitated the discussion about the blueprints of STOEMP. All the “founding” participants (wave 1) felt they could provide their perspective on the mission and goals and that they were listened to. They explained that this bottom-up approach (i.e., the members created the goals and mission together) in the beginning made them feel (and still feel) responsible for the success of STOEMP. During these open discussions, stakeholders gave accounts of becoming aware of numerous existing initiatives taking place in Ghent around the topic of sustainable and healthy food for all. However, these are perceived as being small and fragmented actions with minor successes. This open discussion led to the decision to develop STOEMP as a network bringing people from different organizations and sectors together to raise awareness, inspire, and activate, as well as to connect and strengthen initiatives to make “good food” available to everyone, in particular vulnerable groups.

In the next sections, the factors facilitating and hindering cooperation in the STOEMP network, the successes achieved, the challenges involved in reaching vulnerable groups, and future plans of the STOEMP network will be described in detail.

### 3.3. Facilitating Factors for the STOEMP Networking

#### 3.3.1. Motivation

##### Shared Goals and Concerns

Many study participants stated that the organizations involved in the network had different perspectives and priorities but similar goals and concerns. They acknowledged the social gradient in health and the need to take action to curb this, but also shared their concerns about the complexity and scale of the issue and the ability of the organizations to make a difference. The STOEMP network was seen as crucial by all participants since working together and overcoming organizational, sectoral, and political boundaries could help to tackle this urgent problem and to make an impact. They all felt it was natural for this group of stakeholders, with similar and compatible missions, to come together.

P_wave2_: “We are not going to solve this [limited access to healthy and sustainable food] with our own organization solely thus we need to work together.”P_wave2_: “Everyone has -coming from his/her own job, function or organization- a feeling that we have to be there [in the STOEMP meetings] because we work with vulnerable groups or our work concerns nutrition or health for all. So I think it is actually a logical get together.”P_wave1:_ “[…] what brought us together…the one organization focuses more on one aspect [sustainable nutrition], the other organization more on another aspect [reaching disadvantaged people] but the differences are smaller than what connects us… that is what brought us together.”

##### Perceived Benefits

All study participants highlighted valuing this network due to the benefits they achieve through being involved. Being a member of the STOEMP network was perceived as increasing their knowledge, expanding their network, and thus stretching their reach and impact, enabling new connections or enhancing cooperation between organizations, and becoming inspired and energized. All study participants indicated that they faced the same challenges in making healthy and sustainable food available for all. Study participants mentioned that the opportunity to share experiences, including successes and failures within the network, enabled continuous learning and inspiration, and expanded their horizon, but also created a feeling that they could achieve more by working together in the future. Taking part in STOEMP benefited the stakeholders since it enabled them to share experiences and learn from others, and to pool resources. Study participants also expected STOEMP to benefit target groups by extending the reach and impact of the efforts of all those involved in the network, both directly and more indirectly through policies.

P_wave3_: “There are so many initiatives that are already developed by single organizations but they are so fragmented and often miss their goal because they do not find the right people to work with or it is too much for your own organization. This network helps to share what you have been doing or trying to do and to learn from each other on how to improve, to learn for the future who can be of help when you have new plan, by working together you can achieve more.”P_wave1_: “To me, the connection and networking with other stakeholders broadens my viewpoint. I start to look at things from another perspective because of this cooperation. For me, that is a huge added value of STOEMP. It is increasing my competence.”

##### Ownership

When assessing feelings of ownership among the participants, it appeared that two groups had different experiences within the STOEMP network. The core group (wave 1) that was involved from the start indicated that the mission and goals were developed in a participative co-creation process. As a consequence, these participants still feel very much responsible for the success of STOEMP and support its mission as a network. The second group, that came in later, agreed that actions and plans within STOEMP are always guided by existing needs and perspectives of the group but feel less ownership over it. A few participants from waves 2 and 3 believed there was “distance” or a discrepancy between the goals of STOEMP and the nature of their own tasks within the organization. It was also notable that this second group (wave 2 and 3) questioned the mission and goals of STOEMP as a network as they expected a more action-oriented STOEMP that initiates new projects.

P_wave1_: “The decisions are made based on the input of the group… from the people itself […] to make proposals and create ideas […] they perceive it as their own story, they consider STOEMP as something they created… this facilitates taking their responsibility and supporting the initiative.P_wave2_: “I think the idea of STOEMP grew from the local government… a bit top down… but also bottom-up […] but I don’t feel ownership… I feel that it is a bit further away from my key tasks.”

##### Openness, Time to Build Trust, and Constructive Attitude

Several participants highlighted the open atmosphere and the constructive and respectful interactions within the STOEMP network. A few study participants even underlined that this is not so obvious given they are sometimes working across competing sectors/departments/organizations, i.e., opposing political parties in the municipality, competing health insurance companies. Nevertheless, a few participants mentioned that it took some time to get to know each other and build trust.

P_wave3_: “I think that everyone in STOEMP has a constructive attitude. Everyone wants to contribute where possible and that enables a smooth process. There are also critical comments… but they are always accepted. It even makes us stronger when we can provide an answer or solution.”

##### Personal Values and Enthusiasm

Several participants expressed great enthusiasm for STOEMP out of a personal commitment to social justice and health for all, rather than simply as part of their job. These participants really believe that STOEMP could make a difference in Ghent for its residents in need. 

P_wave1_: “I think that the right people with an open perspective are joining… people with common interests and a support base… people that genuinely think this issue [i.e., making good food available for all] is really important and want to put effort into that.”

#### 3.3.2. Opportunity

##### Face-to-Face Meetings

Several study participants underlined the relevance of face-to-face gatherings within STOEMP as part of the smaller working groups, open meetings, or inspiration days. Meeting the other stakeholders “in person” was seen as facilitating more cooperation.

P_wave3_: “When you cannot put a face on an organization, then it becomes more difficult... it is just when working together face-to-face that you get to know people better … this enhances future cooperations.”

##### Shared Values and Existing Networks

Most study participants worked in organizations with themes and goals that were similar to those of STOEMP. But, as mentioned earlier, STOEMP initiated from two different working groups/networks of relevant stakeholders that were already planning to try and tackle the challenge of making good food accessible for all. The first group originated from the Food Council of Ghent responsible for the local food policy, Ghent en Garde, and focused more on increasing the consumption and production of sustainable foods and creating additional social value. The second group arose from cooperation between the Welfare and Equal opportunities Department of Ghent and the Community Health Centres and the Local Health Promotion organization and primarily targeted healthy foods for all. Participants stated that it felt obvious to bring these two working groups together and create a synergy as soon as they knew of one another’s existence. Moreover, many participants highlighted that “there was a momentum for an initiative like STOEMP.” In addition, several participants mentioned that the STOEMP network grew rather quickly as many stakeholders asked other interested stakeholders with whom they cooperated to join.

P_wave3_: “It [climate change] is hot topic at the moment. I think we all notice that we can do this [put healthy and sustainable food for all on the agenda] and that more and more is happening in the society around sustainability.”P_wave1_: “For me, this was a stroke of luck that we wanted to focus on that challenge at the same time. We, from [name of his/her organization], and then my colleague from [name of her organization] and her team. To look at this from two different perspectives, from two networks actually, that was a strength from the start.”

##### Resources and Political Support

Several study participants mentioned that the municipality provided a budget to support STOEMP at the beginning. However, the downside of this was that the politicians expected a specific media campaign. Some study participants thought this funding could have been used in more efficient ways than running a campaign. However, all study participants finally appreciated the “STOEMP” theme and logo creation and its launch via the media. Nevertheless, all study participants stated that without the financial support (although still seen as limited) the existence and growth of STOEMP would be jeopardized. Study participants explained that this is because the financial support is used to fund both a coordinator, who can put time into managing the STOEMP network and all its events and plans, and an external (neutral) process manager, who facilitates discussion during the meetings. Participants highlighted that both the network coordinator and the external process manager (facilitator) were essential for the success of STOEMP.

Additionally, some participants saw the partnership agreements between the municipality and some organizations within STOEMP (i.e., Community Health Centres, Local Health Promotion organization) as facilitating factors. These agreements represented to participants in the study the commitment of stakeholders to focus on the agreed core tasks. These core tasks were decided through consensus between STOEMP and the municipality. Several participants highlighted that it is unique in their experience to have three administrative sectors (Environment, Health, and Education) run by people from different political parties working so closely together.

#### 3.3.3. Capability

##### Ability to Find the Right Stakeholders and Grow

All study participants from the core team (wave 1) mentioned that it was easy to find the right people to get involved in STOEMP via their networks. This made it possible for STOEMP to grow quite quickly from a core group of about six organizations to more than 20 at the time of this study. 

P_wave1_: “It is also a bit about using your own network to search towards extra stakeholders that can further strengthen the STOEMP network. And I think we managed that quite well, this directed approach… We really reflected about who is still missing, who do we need for what exactly… which actor or target group is not represented yet and who could we involve…”

##### Expertise and Experience of Stakeholders

Several study participants highlighted the relevance of the current stakeholders’ collective ability to advance STOEMP’s goals and objectives. The study participants have different perspectives and tasks within their organizations but all work around this theme (i.e., promotion of healthy and/or sustainable food and/or reaching vulnerable groups) and contribute their specific experiences and areas of expertise. Bringing this knowledge and skills together benefits all stakeholders and their ability to tackle the complex issue at hand. 

P_wave2_: ”You are in a network because you want to share your expertise with others. But also because there is much knowledge that you do not have… I think this is crucial. By working together with different organizations with a different knowhow and background… that is the main reason why this network keeps existing in my view.”

##### External Process Manager

All study participants were unanimous that the inclusion of an external process manager to moderate the general meetings was crucial to the success of STOEMP. This neutral person facilitated discussions between the different organizations and sectors with their sometimes competing interests and ideas. They also emphasized her enthusiasm and expertise in guiding groups, creating a good atmosphere in which everyone felt comfortable to exchange.

P_wave2_: “A meeting where she [external process manager] is present, is really an added value. When she is not there, meetings can be long with a lot of talking and dreaming. But when you leave the meeting, we actually have not made much progress.”P_wave1_: “What is also very important is that we had someone that in an objective manner could counter certain discussion and bring people together. This was not so evident in the beginning so that is why we made this decision in the first place… to have an external person guiding us.”

##### Safe Environment for Critique and Reflection

Several study participants highlighted the open and constructive atmosphere within STOEMP in which certain decisions made earlier could be questioned and approaches could be critically evaluated and thus improved. A few study participants referred to the resilience of this network and the opportunity to reflect on the aspects that did not go well and revise them accordingly.

P_wave3_: “I think that everyone in STOEMP has a constructive attitude. Everyone wants to contribute where possible and that enables a smooth working. There are also critical notes… but they are always accepted. It even makes us stronger when we can provide an answer or solution.”

##### Working Groups around Specific Topics

All study participants valued the creation of the working groups within STOEMP because this makes it possible to target and action the more specific goals of the network. The working groups focus on: (1) Policy impact; (2) data collection and analysis; (3) and communication. Stakeholders can also choose to join the working group that best fits their needs. This made study participants feel “more useful” and able to contribute more to STOEMP.

P_wave1_: “One participant wants to work very specific and action-oriented, another wants to focus on the policy level… it is interesting having these working groups and to see what comes out of them. On the one hand, you still have the process of the whole group with its general strategy. At the other hand, the working groups can target concrete actions and fill in specific needs of people.” 

### 3.4. Barriers for STOEMP Networking

Some study participants (wave 1) highlighted that they feel that for some “newer” stakeholders the goals of STOEMP are a bit vague. This was confirmed by some study participants who had joined the STOEMP network more recently (wave 3). STOEMP’s main objectives are to strengthen participants and organizations, however, some participants feel the need for a more action-oriented network. Furthermore, several participants mentioned that they felt there was too much talking and little action. According to several participants, many meetings were arranged but they felt that the discussions do not always tend to lead to specific actions. Some of the participants in wave 3 did state that in the last general meeting this was picked up and a new discussion about the STOEMP goals and future was started. Some study participants in wave 2 also highlighted that it is important for them to prove to their own organizations that results are being obtained.

P_wave3_: ”I think people see the added value of the network. But personally, I am an impatient type, I feel like… what are we going to do? Because that is the danger with a network, you keep on talking. We could do this and that,… that might be an idea… But if nobody really makes it specific and puts it into an action plan… Inspiring each other is of course important but …”

Several study participants mentioned their uncertainty about the continuation of STOEMP without funding from the municipality. Although some participants mentioned that STOEMP would survive given that so many of those involved in the network feel the need, these participants also acknowledged that the coordinator and process manager play a key role in the current success of the initiative. However, shortly before the local elections (October 2018), participants emphasized that the political support in itself translated more into political demands since politicians wanted to see their input acknowledged (i.e., politicians were demanding that the network delivered what they wanted, so that they could campaign on this basis). Furthermore, several participants emphasized the need to have more impact on policies in the city. They want to ensure that STOEMP’s goals and objectives become integrated more tightly into the work of the municipality, and not just reflected in policies relating to the environment, health, and social affairs but also those relating to other sectors, like the hospitality sector and education. They all highlighted this as a difficult but vital challenge.

P_wave1_: “That is the story of politics. Finding a balance and making sure that the policy is in on it too. Because that is still difficult. Everyone [involved politicians] wants to score. That is a balancing exercise to make sure that everybody gets a piece of the pie.”P_wave1_: “That is why I sometimes doubt… this participative approach like it is happening now, people without formal agreement that put energy into this network. That makes it difficult because when tomorrow the politicians decide to stop the funding, including the time the coordinator puts into STOEMP, then we have nothing anymore…”

The different participants mentioning these issues also indicated that STOEMP really brought organizations closer together and increased trust. Nevertheless, there were participants that remained wary of one another. Some participants highlighted that the cooperation between certain stakeholders was a bit difficult and counter-productive. For example, they acknowledged that it was and sometimes remains difficult for people to lower their guard vis-à-vis the more commercial stakeholders in the network (e.g., health insurance companies) and to work together. A few stakeholders expressed negative attitudes towards each other. A few participants added that it is not always easy to take an integrated approach and unify the different perspectives and tasks of the organizations involved. Additionally, several participants emphasized the lack of involvement of certain stakeholders and sectors such as stakeholders that work closely with the target group (i.e., social restaurants, community centres), or more commercial stakeholders (i.e., hospitality business, local retailers). Nevertheless, participants stated that STOEMP aims to attract these organizations in the future (see section Future/what next).

A few participants of wave 1 mentioned that throughout the year some stakeholders resigned from the STOEMP network but this was mostly due to internal changes in their own organization (e.g., replacement of personnel or organizational modifications), not because they were not satisfied with the work of STOEMP.

Some study participants expressed a decline in their motivation to be involved in STOEMP following the lack of success of the “STOEMP inspiration day” organized in 2019, which attracted fewer people than expected, compared to the event that took place in 2018. This made participants question the relevance of putting their time into this network.

### 3.5. Reaching Socially Vulnerable People via STOEMP

Most study participants highlighted that the current goals of STOEMP are to raise awareness, inspire, and activate, and to connect and strengthen stakeholders to make healthy and sustainable food available for all. The study participants stated that this is what has happened so far. Many relevant organizations have come together to exchange good practices and lessons learned so as to reach vulnerable groups and increase their access to healthy and sustainable foods. However, many participants highlighted that it is too soon to expect a direct impact on the target group. While the exchange of knowledge and expertise has in theory enabled all participating organizations to improve the nature of the services they provide to target groups, it is not yet clear whether these improved services have created positive impacts. Nevertheless, several participants highlighted that in the future it will be vital to further involve people experiencing vulnerability directly in the network to gain additional insight into what can work for them and how.

P_wave1_: “I don’t think we need to pretend that STOEMP already reaches the vulnerable people. I think we are currently at a level where we increase each other’s competence and interest, create a mind shift, … We are at the point where we impart perspectives and share methods and materials to work with our own target group. To unravel what initiatives are available in Ghent is already difficult… Making sure that the possibilities are known for everyone already increases accessibility.”

The study participants also highlighted several other ways in which STOEMP has been perceived as successful. In general, study participants mostly mentioned that STOEMP brought organizations together and facilitated more cooperation. STOEMP also organized two inspiration days (albeit more successful in 2018 than in 2019) to draw attention to STOEMP and its objectives. Furthermore, STOEMP developed a new tool, the ‘STOEMP check list’, as a guide for event developers, the food and hospitality industry, schools, and work places to make more healthy and sustainable choices when providing food. The checklist includes items to support the aims of social inclusion, such as: Are the foods provided affordable?; are foods accessible for different cultures?; is the timing of the event good for all? Another success story was the spread of the ‘Ratatouille’ initiative. ‘Ratatouille’ includes a range of general education and health education sessions for parents living in disadvantaged circumstances. Although this initiative was developed more than 10 years ago, it gained more popularity with the help of STOEMP since several organizations communicated about it. Additionally, STOEMP is currently creating a database with all available initiatives in the city of Ghent that target vulnerable groups and focus on healthy and sustainable foods for all, which might facilitate future implementations of these initiatives.

### 3.6. Future/What Next?

As mentioned earlier, several study participants stated that there is room for improvement of STOEMP. Plans and opportunities were discussed during the interviews. An overview is presented next.

#### 3.6.1. Growth of the Network: Adding New Sectors and/or Stakeholders

Most stakeholders highlighted the need to add more partners from different sectors to the network, especially the for-profit sector. Many felt that the involvement of for-profit organizations could help increase STOEMP’s impact. However, participants also mentioned that they will need to apply different tactics to reach and involve the private sector. A few participants questioned the addition of new stakeholders, and saw more value in strengthening the existing network and increasing its current impact instead of continuing to add new members and discussing the same issues all over again.

P_wave1_: “It would be ideal to also engage some… less typical sectors … it is not so easy. We will have to use different channels to recruit and involve them… chefs of restaurants do not have time to come and sit there brainstorming… but I think it would make STOEMP stronger if we also reach the for-profit sector.”

#### 3.6.2. Impact on the Policy Level

Several study participants highlighted the need to have more impact on policy (e.g., to facilitate structural changes in Ghent to increase access to “good food”). They indicated that politicians are gradually accepting this issue as important, but that progress is rather slow. However, one participant stated that the existence and size of the network and the fact that it reflects the concerns of so many intersectoral stakeholders about access to “good food” puts some pressure on politicians to act on the matter. Some study participants highlighted that it will be crucial for the next phase to focus on creating output (e.g., data on the prevalence of the limited access to healthy and sustainable food in Ghent) to convince politicians.

#### 3.6.3. Restructuring the STOEMP Format and Goals

Some study participants mentioned the necessity for more action-oriented work within STOEMP, although not all participants agreed on this. Other study participants stated the need to be more result-oriented, which links back to the earlier remark on the need to impact policy and convince politicians with scientific data. Reaching and involving the target group directly was also raised as a necessary goal. Currently the network focuses more on reaching them indirectly. One of the participants of wave 3 (the last interview round) mentioned that new specific working groups were being created to address emerging concerns, which focus on the visibility of the current initiatives and how to involve people from the target group. Several participants also suggested a new structure for the STOEMP network, namely to embed it more in different organizations to thus overcome the possible loss of political (financial) support in the future. A few study participants had large ambitions regarding the evolution of STOEMP. In the future, they see STOEMP as the central platform for all organizations in Ghent for questions and support around the theme of improving access to ‘good food’. Additionally, they also see STOEMP as connecting and providing a signaling function amongst different stakeholders, i.e., the local government, academic sector, and intermediary organizations, in relation to this theme. Some study participants even hope it evolves into a kind of certification: i.e., “STOEMP APPROVED” to flag initiatives that have complied with specifically developed STOEMP criteria.

P_wave2_: “If you want to create something sustainable that has an impact then… at a certain moment you will have to start focusing on specific actions.”P_wave2_: “We have to focus on the policy level, we have to wake up the policy makers and start creating big changes.”P_wave1_: “You have to make progress. Whatever progress, do we want to start our own projects or not? It doesn’t matter what we choose but it has to count… It has to create an added value for the stakeholders of the network and convince them to stay involved. Because of the impact of the actions or to learn from it or to change the policy… It can be anything but we have to make sure that stakeholders feel that there is a return-on-investment.”

## 4. Discussion

The evaluation of the inter-sectoral working of the STOEMP network was conducted in the framework of the INHERIT project as STOEMP is one of the first initiatives to place a specific emphasis on what can be done to ensure that city food policy also reaches and benefits the more and most vulnerable segments of a city’s population, and to do so in a systematic way. STOEMP aims to bring together different sectors around common interests as well as to engage people via a co-creation process that defines the goals and plans on what to tackle. This study investigated the process of intersectoral cooperation and the factors that facilitated this (providing participants with the capabilities, opportunities, and motivation to do so), as well as the barriers involved in bringing different actors from different organizations and sectors together to identify what is needed to make healthy, sustainable food available to everyone. The findings can inform and facilitate the creation of similar approaches and networks in other municipalities. At the time of the study, not much could be said about findings in terms of what works best to reach disadvantaged people. The focus of the study was not so much on this, as on the process of bringing together different stakeholders to collectively discuss and identify what is needed. The findings reflect that this can be a fruitful approach, although it is also a challenging one.

STOEMP evolved through a process whereby different sectors realized they had similar goals that they could solve more efficiently and effectively by collaborating and co-creating solutions, which provides it with strong foundations. The rapid growth of the STOEMP network, originating from the cooperation of two different departments and nowadays including more than 20 organizations, has been partly attributed to the shared goals and concerns of different stakeholders within Ghent that wanted to tackle the inequality in access to healthy and sustainable foods. Finding the right stakeholders that have a similar mission is crucial to make a good start. Research results indicated that the study participants were very motivated to be involved in STOEMP. The Self-Determination Theory [32] suggests that when the needs for competence (to gain knowledge and skills), connection (to experience relatedness to other people), and autonomy (to feel in control) are fulfilled people feel a more intrinsic motivation which will translate into more commitment, passion, interest, and satisfaction. All study participants highlighted a range of benefits of their participation in the STOEMP network, including an increase in their expertise and skills, a wider network to collaborate with, and more inspiration and energy to tackle the complex issue of ‘’good’’ food for all. Moreover, the bottom-up process (i.e., members co-constructed goals, visions, and solutions) of STOEMP created feelings of ownership in most participants. The STOEMP network seems to cover all necessary needs of Deci and Ryan’s theory [32], which could explain the maintained high motivation and enthusiasm in the network and the limited drop-out of organizations involved. Some of the difficulties identified, or barriers with respect to cooperation are uncertainty about continuation and funding, political demands, difficult and counter-productive cooperation between stakeholders, and a lack of involvement of certain stakeholders and sectors in the network. The network appears to be struggling somewhat to identify concrete actions needed to achieve the real structural changes required to improve access to sustainably produced, healthy food among vulnerable groups. This has led to a sense amongst some study participants that it is not action-oriented enough, and the motivation of some participants is faltering. The difficulty of finding ‘solutions’ is not surprising, given the complexity of the issues at hand. Food systems have evolved to produce large quantities of food, but the focus has not, simultaneously, been on the production of healthy and environmentally sustainable food [3,4,26,33]. As a result, where people with fewer resources have access to food, it is often cheap, highly processed and of poor nutritional quality [26]. Enabling the structural shifts needed to reorient food systems so that the sustainable, healthier options are the more affordable options is a challenging, long term task that requires coordinated action across a wide range of sectors working across policy levels [19,26]. All actors in the food supply chain (i.e., including consumers, retailers, food industry, and the hospitality sector) should be working together to achieve this [3]. Nevertheless, cooperation between public and private sectors can be challenging. INHERIT’s findings from an analysis of intersectoral cooperation in 12 different kinds of initiatives around Europe found that cooperation between sectors, including between public and private sectors, is related to motivation connected with perceived mutual benefits, shared common goals, the added value of cooperating, and the need to cooperate [34]. Additionally, INHERIT conducted a roundtable with 18 representatives from leading businesses (i.e., including the food industry e.g., Colruyt and Unilever) and senior European commission officials. The discussions reflected that many private sector actors want to, and are taking actions to promote the production and consumption of healthy and sustainable food. However, they need the public sector to regulate this to ensure a level playing field, and consumers to support this change [35]. STOEMP is also inherently dealing with the complex challenges involved in addressing vulnerability, and accessing vulnerable groups. Beyond the fact that ‘good’ food may not be affordable, many people may not want to consume these foods for cultural reasons, or simply because they are not used to, do not see the benefits of, and do not like eating them [9]. When brought together, the barriers involved in making sustainable and healthier foods more widely affordable and accessible in general, paired with the barriers of ensuring these foods gets to vulnerable communities, are considerable.

STOEMP provides a mechanism to enable cooperation across sectors, which has been identified as key to tackling complex societal issues, in particular the wider social determinants of health and health equity [25]. Working alone, the environment, health, and social sectors can only address ‘symptoms’ of the wider structural causes that make it so difficult for vulnerable groups to access ‘good’ food [19,26]. The Food Savers initiative for example, has been very successful in redistributing unused food from supermarkets to vulnerable groups. However it operates on the basis of a system that is malfunctioning, in that fresh, good quality food is being produced but many people do not have the resources to buy it, and buy lower priced calorie dense food products. Cooperative organizations like STOEMP enable people to collectively think through the systems and behaviors that are ‘driving’ the status quo, and what can be done, both to alleviate immediate difficulties people face but also to change the underlying problems and barriers.

The findings from INHERIT on ‘‘elements of good practice’’ [26] as well as the policy recommendations (https://inherit.eu/policy-toolkit/) from the INHERIT project can serve as a guidance on how to address some of the barriers mentioned earlier. Anchoring local initiatives in higher level priorities at a national or European level can stimulate broader policy change (e.g., create support for a shift towards a food policy on the European level) and secure higher level political support. It is crucial that these policies have the right focus, which has not traditionally been the case. The imperative of food systems has been on quantity, rather than quantity and quality for all [26,36]. Organizations involved in STOEMP can remind policy makers of the importance of ‘good’ food for all as a legitimate and important overriding goal, in the context of the climate crisis and in addressing health inequities. They must demonstrate how their approach can help deliver solutions. To do this, the STOEMP network is currently conducting evaluations of its initiatives as well as collecting data on the limited access of ‘’good’’ food for disadvantaged citizens to convince policy makers of the necessity of STOEMP and its impact. This also aligns with the INHERIT recommendation [26] to integrate qualitative and quantitative evaluations of ‘triple-win’ initiatives as this can advance an understanding of implementation processes, intersectoral cooperation, impacts, and benefits, as well as provides insight in what could be done better to build synergies across sectors and to enhance outcomes. Moreover, Blay-Palmer and colleagues [37] underlined the benefits of sharing good practices across (global) communities thus enabling communities to adopt good practices appropriate for their unique context. According to Blay-Palmer and colleagues [37], “a global network of community/regional food initiatives that share solutions to common global pressures could help to counter the influence of the industrial food system as it could enhance the sustainability and resilience of food systems by linking regional initiatives into a global network where global challenges and successful solutions continue to build global solidarity.” STOEMP is currently creating an overview of the good practices in Ghent. This catalogue may be useful to other community and regional governments in Europe, who may take similar steps. STOEMP has been funded and supported by three administrative sectors of the local government (Environment, Health, and Education) and run by people with different political affiliations. This funding and support made it possible for STOEMP to grow to its current form. However, there is the risk that if funding drops, STOEMP may vanish. The recent publication of the (preliminary) coalition agreement of the city indicated a decrease in the budget allocated to the food policy of Ghent. This underlines the importance of searching for alternative funding sources, for example a hybrid business model with diverse funding partners across sectors. Some study participants envisioned the incorporation of STOEMP in the organizational structure of the involved organizations as another solution.

This is the first qualitative evaluation of an intersectoral initiative that aims to identify and implement more effective ways to provide ‘good’ food for all. One of the strengths of this study was the involvement of a diverse group of stakeholders, whose insights, when taken together, provide a broad and comprehensive account of how STOEMP works. Data saturation was reached given that the different participants highlighted similar facilitating and hindering factors and envisioned shared future plans and current limitations. In addition, the methodological approach, including the step-wise and iterative process of data collection and analysis together with the application of the principles of Appreciative Inquiry [29] and the COM-B model [27], as well as the inclusion of two raters to code and analyze the data increased the reliability and validity of our findings. This study also had a few limitations. Although we reached data saturation, it is possible that our qualitative analysis did not capture all relevant factors facilitating and hindering the STOEMP intersectoral network in its efforts to achieve its goals. Other qualitative methods such as observation of the open meetings, inspiration events, and working groups could have given additional insights into STOEMP. Moreover, given that we were not able to interview STOEMP stakeholders that resigned and only included current STOEMP members, our findings might be biased because of the more positive opinions of the (enthusiastic) participants. However, the fact that most participants did make critical remarks in the interest of improving STOEMPs ability to achieve its objectives makes this seem unlikely. In addition, contextual factors specific to each city might affect transferability of the STOEMP approach. Ghent has a favorable political, social, and economic environment for such initiatives since it made an early decision to become one of the first European cities to launch its own urban food policy, called Ghent en Garde in 2013. Nevertheless, other municipalities could learn much from the facilitators and barriers of the STOEMP initiative.

## 5. Conclusions

The current study investigated the process of intersectoral cooperation in the STOEMP network, which brings together representatives from a range of organizations operating at the municipal level to identify how to make healthy, sustainable food available to everyone. Opportunities highlighted and barriers faced reflect the challenges, strengths, and weaknesses of taking a participatory, systems-based approach to identifying and taking effective action to widen access to healthy and sustainable foods. STOEMP would like to influence policy and thereby strengthen its impact, but is yet to formulate collectively how this can be done. An investigation into the actual impacts of STOEMP on vulnerable people and policy related activities and recommendations can be the subject of future research, once the network has evolved further. A consistent ‘’triple-win’’ focus can help the network remain oriented towards finding effective solutions to integrate health, environmental, and social challenges. The recommendations from the INHERIT project, including the Policy Tool Kit, could help STOEMP consider further structural changes needed to ensure that the ‘’Ghent en Garde’’ city food policy achieves its objective of making ‘’good’’ food available to all, including its more and most disadvantaged populations.

## Figures and Tables

**Table 1 ijerph-17-03073-t001:** Code book with theoretically deduced codes and inductively created themes.

Codes (Theoretical Deduction)	Themes (Induction)
Initiation and development of STOEMP (Section 3.2.)	Trust
	Fine-tune mission and goals
	External ‘’neutral’’ process supervision
	Bottom-up approach
Facilitating factors (Section 3.3.)	
Motivation (Section 3.3.1.)	Shared goals and concerns
	Perceived benefits
	Ownership
	Openness, time to build trust and constructive attitude
	Personal values
Opportunity (Section 3.3.2.)	Face-to-face meetings
	Shared values and existing networks
	Resources and political support
Capability (Section 3.3.3.)	Ability to find the right stakeholders and grow
	Expertise and experience of stakeholders
	External process manager
	Safe environment for critique and reflection
	Working groups around specific topics
Barriers (Section 3.4.)	Need for action-oriented working
	Uncertainty about continuation
	Funding
	Political demands
	Difficult and counter-productive cooperation
	Lack of involvement
	Resignation of STOEMP
	Decrease in motivation
Reaching socially vulnerable people (Section 3.5.)	Too soon to expect a direct impact
	Success stories
Future/what next? (Section 3.6.)	Growth of the network
	Impact on policy level
	Restructuring the STOEMP format and goals

**Table 2 ijerph-17-03073-t002:** Characteristics of the study participants (*n* = 15).

	Participant Number	Organization	Sector	Time Involved in STOEMP
Wave 1 (Core team)	1	Municipality, Department of Welfare and Equal Opportunities	Health	From the start
	2	Municipality, Department of Urban Planning	Environment	From the start
	3	Community Health Center	Health	From the start
	4	Local Health Promotion Organization	Health	From the start
	5	External (neutral) process manager	/	From the start
Wave 2	6	Non-profit organization promoting a vegetarian lifestyle	Environment	1 year
	7	Municipality, Department Education	Education	6 months
	8	Health Insurance company 1	Health	1.5 year
	9	Municipality, Department Education	Education	1 year
	10	Health Insurance company 2	Health	2 year
Wave 3	11	Teacher Secondary School	Education	1 year
	12	Social grocery store	Health, Environment, Retail, Social Economy	7 months
	13	Local sustainable restaurant	Hospitality	1.5 year
	14	Social entrepreneur	Health, Social Economy	1 year
	15	NGO	Environment, Health	3 months

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
