# Peer review of "Qualitative Evaluation of the STOEMP Network in Ghent: An Intersectoral Approach to Make Healthy and Sustainable Food Available to All"

_ijerph, 2020, doi:10.3390/ijerph17093073_

Round 1

Reviewer 1 Report

Thank you for the opportunity to re-review this paper. I thank the authors for replying to my original and subsequent comments and I believe it has strengthened the paper in respect of identifying the participant by wave (in the absence of a unique identifier due to small participant numbers). Additionally, I believe it is most helpful when the participant coding indicates that the themes transcend waves (eg) Research benefits (compared and contrasted with the Shared goals and concerns section). For the Shared goals and concerns section, a Wave 1 quote (involved from the start cohort, if available) would be useful to have. The same applies re: Restructuring the STOEMP format and goals.

It was definitely helpful to understand differences by waves. Thank you for acting on this feedback to strengthen the paper.

Some additional quantification of the qualitative analysis would be helpful (eg) how many participants equate to 'several', 'some', and 'a few'? Some understanding of number of participants in accordance would be welcome.

Page 12, paragraph 3 - delete redundant comma: "...that took place in 2018."

Page 14, line 7 - delete 'a': "...facilitate the creation of similar approaches and networks..."

Page 16, line 7 - spelling error re: 'However'

Page 16, lines 2-3 of the conclusion - "... brings together representatives from..." 

Author Response

***Reviewer 1

Thank you for the opportunity to re-review this paper. I thank the authors for replying to my original and subsequent comments and I believe it has strengthened the paper in respect of identifying the participant by wave (in the absence of a unique identifier due to small participant numbers). Additionally, I believe it is most helpful when the participant coding indicates that the themes transcend waves (eg) Research benefits (compared and contrasted with the Shared goals and concerns section). For the Shared goals and concerns section, a Wave 1 quote (involved from the start cohort, if available) would be useful to have. The same applies re: Restructuring the STOEMP format and goals.

Answer: We added quotes from Wave 1 to both sections.

It was definitely helpful to understand differences by waves. Thank you for acting on this feedback to strengthen the paper. Some additional quantification of the qualitative analysis would be helpful (eg) how many participants equate to 'several', 'some', and 'a few'? Some understanding of number of participants in accordance would be welcome.

Answer: We thank the reviewer for the comment. However, given the qualitative character of this study, we do not believe it is appropriate to include the exact quantification as the data analysis is based on the interpretation of the study participants’ words by the involved researchers and the reported findings are not quantitative research per se. In general, we used ‘a few’ and ‘some’ when only 1-3 study participants (out of 15) mentioned a certain (similar) statement, ‘several’ was used when 5-7 study participants stated this, and ‘majority’/‘most’ was used when respectively more than half/ almost all study participants highlighted a certain aspect.

Page 12, paragraph 3 - delete redundant comma: "...that took place in 2018."

Page 14, line 7 - delete 'a': "...facilitate the creation of similar approaches and networks..."

Page 16, line 7 - spelling error re: 'However'

Page 16, lines 2-3 of the conclusion - "... brings together representatives from..."

Answer: These errors have been corrected.

Reviewer 2 Report

This is a very interesting paper and it makes a valuable contribution to our understanding of how to develop and implement multi-sectoral interventions to promote healthy and sustainable eating in community contexts.  It also resonates with the growing interest in the use of qualitative methods in evaluation and the importance of understanding the how of interventions as well as measuring impact.  The paper is clearly written and I enjoyed reading it.  To enhance its value there are a few points that could be elaborated.

Introduction

  • Could you explain a bit more what you mean by a systems approach? Does this mean looking at the food system and environment as a whole?  I am assuming not systems theory.
  • And related to this, which sectors are relevant in this context and included in the STOEMP initiative? In the methods the different types of stakeholders involved are described, but it would be useful to set out here what the relevant sectors are in this context and that need to be brought together. 
  • Defining a sustainable diet is indeed difficult, but was any particular definition used as a target in STOEMP? The Eat Lancet guidelines are cited, but these are quite and extreme and have provoked some controversy.

Methods

  • A minor point, but what kind of purposive sampling?

Discussion

  • While the focus of the paper is the process of inter-sectoral collaboration and how to facilitate this, it would be useful to have some sense of whether STOEMP is considered to be a success or not and what has been achieved. In the results section you allude to this and that it is perceived to be a success.  Even though you state the impact on reaching disadvantaged social is uncertain, some indication of what has been achieved would help to underscore the value of your findings for others planning such interventions.
  • The range of stakeholders involved in the STOEMP network described earlier in the introduction and methods is actually quite narrow and here you return to the need to restructure food systems, so are any sectors and/or stakeholders that it would be valuable to include?
  • Related to this, it would be interesting here to touch on what promotes collaboration across the private and public sectors. You have some findings on this and crossing this divide is an important one.
  • Do you consider that there are any limitations to your findings? Are there any local contextual factors that might limit the transferability of your findings to other contexts?

Author Response

***Reviewer 2

This is a very interesting paper and it makes a valuable contribution to our understanding of how to develop and implement multi-sectoral interventions to promote healthy and sustainable eating in community contexts. It also resonates with the growing interest in the use of qualitative methods in evaluation and the importance of understanding the how of interventions as well as measuring impact. The paper is clearly written and I enjoyed reading it. To enhance its value there are a few points that could be elaborated.

Introduction

Could you explain a bit more what you mean by a systems approach? Does this mean looking at the food system and environment as a whole?  I am assuming not systems theory. And related to this, which sectors are relevant in this context and included in the STOEMP initiative? In the methods the different types of stakeholders involved are described, but it would be useful to set out here what the relevant sectors are in this context and that need to be brought together.

Answer: We indeed meant the food system and environment as a whole, in which a range of governments (both local and national), organisations across different sectors and individuals need to work together in a coordinated way to make sustainable and healthy foods available for all. We added the following paragraph to the introduction:

“A transformation of the food system should ultimately involve multiple stakeholders, from individual consumers to policy makers and all actors in the food supply chain, working together towards the shared goal of healthy and sustainable diets for all.” [3]

Defining a sustainable diet is indeed difficult, but was any particular definition used as a target in STOEMP? The Eat Lancet guidelines are cited, but these are quite and extreme and have provoked some controversy.

Answer: We added the following paragraph to the method section.

for the environment, sufficient for all, tasty, and fair.”

Methods

A minor point, but what kind of purposive sampling?

Answer: We added the kind of purposive sampling of the first two waves in the method, i.e. heterogeneity or maximum variation. We tried as much as possible to select participants from different sectors with expected varying perspectives on STOEMP.

Discussion

While the focus of the paper is the process of inter-sectoral collaboration and how to facilitate this, it would be useful to have some sense of whether STOEMP is considered to be a success or not and what has been achieved. In the results section you allude to this and that it is perceived to be a success.  Even though you state the impact on reaching disadvantaged social is uncertain, some indication of what has been achieved would help to underscore the value of your findings for others planning such interventions.

Answer: Although we would be very happy to highlight the success of STOEMP, at this stage we do not have any numbers to support such a statement. As the reviewer highlights, we do indicate some success stories in both the result and discussion section (e.g. the project Food savers, Ratatouille) based on the interviews with the study participants but we do not have more data than the perceived successes by the interviewees. In addition, the STOEMP network is currently conducting evaluations of its initiatives as well as collecting data on the limited access of ‘good’ food for disadvantaged citizens so we hope that there will be more evidence in the future about this network and its impact.

The range of stakeholders involved in the STOEMP network described earlier in the introduction and methods is actually quite narrow and here you return to the need to restructure food systems, so are any sectors and/or stakeholders that it would be valuable to include? Related to this, it would be interesting here to touch on what promotes collaboration across the private and public sectors. You have some findings on this and crossing this divide is an important one.

Answer: The following paragraphs were added to the discussion:

‘All actors in the food supply chain (i.e. including consumers, retailers, food industry, hospitality sector) should be working together to achieve this [3].’

Do you consider that there are any limitations to your findings? Are there any local contextual factors that might limit the transferability of your findings to other contexts?

Answer: We added the following paragraph in the discussion to the limitations.

Reviewer 3 Report

This paper provides insights into an intersectorial initiative from the perspective of participants who work in member organizations. Such intersectorial work has been called for since the initiation of the Ottawa Charter in 1986 and this paper offers insights into one focused on healthy and sustainable food especially to disadvantaged groups.

The following comments are offered as a way to improve the paper:

  • The grammar and syntax needs to be resolved. For example there is slippage between past and present tense; missing or incorrect propositions; and there is inconsistent or inappropriate use of singular and plural nouns.
  • The opening paragraph uses a reference to make a claim about global health problems. However the referenced paper is focused on obesity and overweight conditions and not ‘other non-communicative diseases’. To support the claim the authors make about global health problems then a more relevant reference is needed. Some ideas –

Frenk J, Moon S. Governance challenges in global health. New England Journal of Medicine. 2013 Mar 7;368(10):936-42.

Ollila E. Global health priorities–priorities of the wealthy?. Globalization and health. 2005 Dec;1(1):6.

Warwick-Booth L, Cross R. Global health studies: A social determinants perspective. John Wiley & Sons; 2018 May 21.

  • Page three and subsequent pages have text using the term ‘good’ food. This is has moralistic overtones and its meaning is implied. The title refers to ‘healthy and sustainable food’ so it this what the author are referring to when they say ‘good’. Suggest that the word ‘good’ is removed and replace with what the authors actually mean.
  • Appreciative inquiry (AI) is mentioned in the data collection however is not referred to later in the articles. The success factors and future developments are sufficiently implied in the discussion and conclusion however explicit reference to AI is needed.

Author Response

***Reviewer 3

This paper provides insights into an intersectorial initiative from the perspective of participants who work in member organizations. Such intersectorial work has been called for since the initiation of the Ottawa Charter in 1986 and this paper offers insights into one focused on healthy and sustainable food especially to disadvantaged groups.

The following comments are offered as a way to improve the paper:

The grammar and syntax needs to be resolved. For example there is slippage between past and present tense; missing or incorrect propositions; and there is inconsistent or inappropriate use of singular and plural nouns.

Answer: This paper has been double checked by the native co-authors of this paper.

The opening paragraph uses a reference to make a claim about global health problems. However the referenced paper is focused on obesity and overweight conditions and not ‘other non-communicative diseases’. To support the claim the authors make about global health problems then a more relevant reference is needed. Some ideas –

Frenk J, Moon S. Governance challenges in global health. New England Journal of Medicine. 2013 Mar 7;368(10):936-42.

Ollila E. Global health priorities–priorities of the wealthy?. Globalization and health. 2005 Dec;1(1):6.

Warwick-Booth L, Cross R. Global health studies: A social determinants perspective. John Wiley & Sons; 2018 May 21.

Answer: We thank the reviewer for the suggestions and selected the first paper (Frenk & Moon 2013) to add additional evidence.

Page three and subsequent pages have text using the term ‘good’ food. This is has moralistic overtones and its meaning is implied. The title refers to ‘healthy and sustainable food’ so it this what the author are referring to when they say ‘good’. Suggest that the word ‘good’ is removed and replace with what the authors actually mean.

Answer: We understand the consideration of the reviewer. However, STOEMP was launched in 2016 as a network of ‘good food’ initiatives taking place in Ghent. The network defined good food as healthy, nutritious, local, produced with respect for the environment, sufficient for all, tasty, and fair. So when study participants were talking about ‘good food’ in the interviews, they referred to STOEMP’s meaning of ‘good food’. Therefore, we think it is best to keep good food but it is also the reason we have put good in between ‘’.

However, we also added the following paragraph in the method section to further clarify:

“ for the environment, sufficient for all, tasty, and fair.”

Appreciative inquiry (AI) is mentioned in the data collection however is not referred to later in the articles. The success factors and future developments are sufficiently implied in the discussion and conclusion however explicit reference to AI is needed.

Answer: We added a reference to the AI in the description of the strengths of the study in the Discussion section.

This manuscript is a resubmission of an earlier submission. The following is a list of the peer review reports and author responses from that submission.

Round 1

Reviewer 1 Report

Thank you for the opportunity to review this interesting study. The authors set a good context in the background / introduction. However, the most up to date data should be used where possible (eg) more recent than a 2014 list of WHO strategic heath cities to ensure the most meaningful context.

Methodological section: Cite the number of interviewees (n=5) for each wave. Were reasons for non-participants' disinterest given?

Reaching socially vulnerable people via STOEMP: The authors explore a lot of content in this section but it reads as a list of activities rather than an analysis of what was useful to reach hard to reach vulnerable groups. More critical analysis would strengthen the paper.

My main feedback is that it would be useful and interesting (and would strengthen the paper) to provide an analysis that differentiates between core and non-core stakeholders' perspectives to provide a more critical evaluation of the enablers and barriers to the STOEMP network and its attainment of its priority objectives. Aligned to this, the identification of interviewees (Px / Py) suggests only two were quoted verbatim throughout. It would be prudent and helpful to provide unique identifiers and affiliate each quoted participant to a stakeholder background, differentiated by their core / non-core status for total openness and transparency. In addition, illustrative quotes for each section would further strengthen the paper and bring the themes to life.

Minor additional proofreading is required before resubmitting.

Reviewer 2 Report

Generic comment

0. As lines are not numbered, I have identified them by the page and section of the paper and the first word of the line, i.e.

Line “opportunities”:

1. There is no needs for a comma after e.g. and i.e. I have mentioned below the two first occurrences of “e.g.,” and the first occurrence of “i.e.,” only.

2. Can the acronym STOEMP be spelled out or translated for the benefit of the curious reader?

3. The paper is very vague about methodological details, the actual size of the STOEMP network, the definition of “stakeholder”, the selection of the sample etc. This is a major weakness.

Another big weakness is the top-down approach where the organisers are assumed to judge how succesful their actions have been without even consulting the “target population”. If I understand correctly, the "target" includes the general public, people who eat in vegetarian restaurants, as well as “vulnerable groups”, which are not better defined.

Page 1
Abstract
Line “opportunities”: e.g., external → e.g. external
Line “ownership”: e.g., time issues → e.g. time issues

The rather long sentence from “The facilitating ...” to “… sustainable food” would be clearer and more elegant if reformulated as

The highlighted facilitating factors and opportunities (e.g. external facilitator, shared values, engagement and enthusiasm, resources, ownership) as well as the barriers faced (e.g. time issues, uncertainty on continuation and funding, discrepancy in visions) reflect the challenges, strengths and weaknesses of taking a systems approach to trying to formulate solutions to widening access to healthy and sustainable foods.

Line “impact”:
Comment: end the sentence after “exact needs.”

Page 1
Introduction
Line “patterns”: i.e., a high intake → i.e. a high intake

Line “ecosystems”: lead to → contribute to

Line “detrimental”
Comment: there is no “urgent need” , and it is certainly not universally perceived as such. I suggest that “urgent need” should be replaced by something like “environmental obligation” or “moral need”, or “moral pressure” or some similar wording. Or you could simply replace “there is an urgent need to transition” by “it is desirable to transition” or “it is in order to transition.”

Line “example”: habitual by nature with relatively → habitually done with

Page 2
Line “showed”
Comment: I have difficulties understanding the sentence “most population-based health promotion efforts result in a worsening of social inequalities in health”, maybe because I do not understand the precise meaning of “population-based health promotion”. There are many examples of successful promotion of reproductive health, improved food storage, water conservation, sanitation etc. in developing countries. I am not so familiar with developed countries but I do not see why advertising affordable good practices should result in more social inequality. In fact, the later part of the paragraph seem to contradict the second sentence.

Line “cities”
Comment: what are “downstream actions” and “upstream policies”? Are you saying that grass-roots actions are not supported by policies?

Line “tend”: actions [17, 18], these authors also acknowledged → actions, [17] and [18] acknowledged
Comment: my guess

Line “dietary”: Furthermore, Pons-Vigues and colleagues [18] argued, based on several reviews and WHO reports [20 – 23] that to tackle the inequalities in health, an intersectoral approach, in which all community sectors are involved, that addresses social and environmental inequalities is required and essential, and more successful than curbing health inequalities from the health sector only
Comment: quite a sentence! Can you simplify this into two sentences?

Line “Vigues”: reports [20 – 23] that → reports [20 – 23], that

Method
Study design and sampling

Line “authors”: March-July → March and July

Line “authors” and (Page 3) “participating”
Comment: are stakeholders beneficiaries? Apparently not. It seems that “stakeholders” are the “organisers” of the network. Stakeholders are normally all those involved, thus including beneficiaries of the STOEMP network. The wording “participating stakeholders” is also somewhat puzzling. Are there non-participating stakeholders? As also underlined in my comment below, you should be more explicit about what you call a “stakeholder”.

Page 3

Line “the study”
Comment about the sentence: “In total 25 (ex-) stakeholders of STOEMP were invited, of which 16 participated, one interview was discarded due to participant’s lack of knowledge about the STOEMP network. “ Does the person who did not know about STOEMP qualify as a stakeholder? Why did you invite 25 people? How did you select them? How many potential “stakeholders” are there? 25? 100? 200?

Description of the STOEMP network

Line “STOEMP was launched”:
Comment: a very clear paragraph, where it appears that a major stakeholder is the general public. Up to this section, your paper gives the impression that STOEMP is taking care of needy people such as the homeless. Your links also suggest that restaurants and businesses and the municipality of Ghent are among the main participants. Are they your “stakeholders” ? Some lines on (Page 3, Line “evolved” under Data Collection) you do however mention “vulnerable groups”. The “stakeholder” group keeps growing...

Data collection

Line “Interviews”
Comment: now you tell us that there are “organisations involved”. Are you stakeholders individuals or institutions, or both?

Line “asked”
Comment: age and gender? If a “restaurant” is being interviewed, I suppose this is the age and gender of the respondent?

Page 4
Data analysis

Page 5
Results

Line “Table 2”: were → had been

Line “in STOEMP”
Comment: 25 stakeholders, minus one who was dropped, is 24. We are left with 11 men and 13 women! Strictly, “most” were women!

Line “(called”
Comment: you have already mentioned this on page 3

Line “(non)-profit”
Comment 1: what is the difference between (non)-profit and non-profit?
Comment 2: “decided that they wanted to invest further in one of their strategic goals, namely creating more socially added value because they felt more efforts were needed to realise this specific goal”. The sentence is redundant: they wanted it because they “liked” it. Suggestion: drop the end of the sentence from “because” to the end.

Facilitating factors for the STOEMP networking
Motivation
Shared goals and concerns

Page 6
Perceived benefits

Line “Also created”
Comment: “Taking part in STOEMP benefited the participants”. OK, but did it also benefit the “vulnerable groups” mentioned above?

Pages 7 to 11: skipped

Page 12
Discussion

Line “of initiatives”:
Comment: I am puzzled by “government”. Up to here I saw only the municipality mentioned.

Line “options”:
Comment: “challenging” and possibly incompatible with “low cost”

Page 14
Comment to “our qualitative analysis did not capture all relevant factors influencing the successes or failures of the STOEMP intersectoral network”. Indeed: There are two groups, the general public and “vulnerable groups” that should have been consulted to decide whether the initiative is successful.

Conclusion

Line “network”: network to reach to make healthy → network to make healthy

Round 2

Reviewer 1 Report

Thank you for the opportunity to review this revised manuscript.

I welcome the additional explanation of STOEMP throughout.

Page 3 Study design and sampling, paragraph 1 - delete redundant use of yet (last word of the paragraph)

Page 4 paragraph 2, fourth last line should read as 'from which the participants were coming'

I could not locate the attachment containing the authors' response to my first review but I note that this revision still does not explain, in the method section, the attribution of Px and Py. Quotations from participants remain unaffiliated. I understand the small sample size limitations but even some differentiation by wave (if not affiliation) may be interesting in further understanding the results. While I note you state (in the discussion) that different participants highlighted similar factors, I would still like to see some differentiation between waves and participants' backgrounds. In this way, I believe that while the research is interesting, it is not optimally reported.

Some sub themes remain unsupported by accompanying illustrative quotes (eg) Personal values and enthusiasm, and Working groups around specific topics

Minor additional proofreading is required:

Page 2, paragraph 2, line 3 should read as promote not promoted

Page 2, paragraph 2, line 4 should read as cities'

Page 2, paragraph 2, line 11 should read as [17 and 18]

Page 7, paragraph 1 - review grammar of last sentence before the verbatim quotes: target groups and targets groups

Page 7 Ownership paragraph, fourth last line - from waves 2 and 3

Page 13, third last line, review grammar of ", as and provide insights"

Reviewer 2 Report

No additional comments

R